# A Molecular Dynamics Study of Monomer Melt Properties of Cyanate Ester Monomer Melt Properties

**DOI:** 10.3390/polym14061219

**Published:** 2022-03-17

**Authors:** Rebecca T. Haber, Andrea R. Browning, Bayleigh R. Graves, William P. Davis, Jeffrey S. Wiggins

**Affiliations:** 1School of Polymer Science and Engineering, University of Southern Mississippi, 118 College Dr, #5050, Hattiesburg, MS 39406, USA; rebecca.haber@usm.edu (R.T.H.); bayleigh.graves@usm.edu (B.R.G.); 2Schrödinger, Inc., Portland, OR 97204, USA; andrea.browning@schrodinger.com; 3Department of Mathematics, Western Washington University, 516 High St., Bellingham, WA 98225, USA; davisw9@wwu.edu

**Keywords:** molecular simulation, molecular dynamics, melting temperature, cyanate ester, monomer melt

## Abstract

The objective of this work was to computationally predict the melting temperature and melt properties of thermosetting monomers used in aerospace applications. In this study, we applied an existing voids method by Solca. to examine four cyanate ester monomers with a wide range of melting temperatures. Voids were introduced into some simulations by removal of molecules from lattice positions to lower the free-energy barrier to melting to directly simulate the transition from a stable crystal to amorphous solid and capture the melting temperature. We validated model predictions by comparing melting temperature against previously reported literature values. Additionally, the torsion and orientational order parameters were used to examine the monomers’ freedom of motion to investigate structure–property relationships. Ultimately, the voids method provided reasonable estimates of melting temperature while the torsion and order parameter analysis provided insight into sources of the differing melt properties between the thermosetting monomers. As a whole, the results shed light on how freedom of molecular motions in the monomer melt state may affect melting temperature and can be utilized to inspire the development of thermosetting monomers with optimal monomer melt properties for demanding applications.

## 1. Introduction

Cyanate esters that polymerize into polycyanurates have excellent processability, low water uptake, low dielectric loss, and excellent thermal stability [1]. These thermosetting polymers occupy a niche area between high-glass-transition-temperature bismaleimides and tetrafunctional epoxy resins [2]. In this respect, they have been utilized for applications such as filament wound Carbon-Fiber-Reinforced Polymers (CFRPs) in heat shields used for atmospheric reentry and airframes [3]. The combination of high performance with good processability is a unique asset of polycyanurates compared to alternative high-temperature thermosetting resins. Research and development of these polymer matrices in the aerospace and defense industries can profit from the production of novel advanced thermosets and a better understanding of the underlying dynamics of these systems at an atomistic level. Accordingly, examining industrially relevant cyanate ester thermosetting monomers with Molecular Dynamics (MD) simulation is of particular interest in this study.

The melting temperature (*T_m_*) of cyanate ester monomers is a key parameter in the fabrication of CFRPs with techniques such as filament winding and resin transfer molding [4]. Optimally, the cyanate ester exhibits a *T_m_* slightly above room temperature (315–340 K) yet is readily solidified for storage [5]. High *T_m_* (above 400 K) makes it difficult to maintain a monomer melt for adequate processing times without advancing the polymerization. Generally, measurement of monomers by differential scanning calorimetry is a common way to quantify *T_m_* and indicate candidates for good processability. However, the main experimental difficulty is in the prohibitively expensive and time-consuming synthesis to not only presuppose the influence of chemical design but also obtain high-purity monomer samples, which is not trivial and requires specialized synthetic training. Expedient computational prediction of *T_m_* could substantially facilitate the thermosetting polymer selection process and inspire the molecular design and synthesis of new cyanate esters for CFRP applications.

Atomistic MD simulations of polycyanurate monomer melt properties have primarily focused on wettability [6] and energetic rotational barriers [5,7] to understand structure–property relationships. Ghiassi. studied the energetic rotational barriers for a series of aryl cyanate esters and demonstrated control of monomer *T_m_* through manipulation of the entropy of melting by substituting silicon atoms for carbon atoms [5]. A further understanding of chemical structure effects can be gained from theoretical *T_m_* calculations for monomers. Another recent work, by Harvey., examined propyl-bridged di(cyanate ester) monomers, and semiempirical models indicated bulky groups ortho to the bridge connection on the phenyl rings inhibit molecular motion in the melt, thereby increasing *T_m_* [7]. While polymers and composites based on cyanate esters have expanded to a plethora of applications, little systematic work has been carried out to examine property prediction of these thermosetting monomers [6].

*T_m_* can be readily calculated from simulations via direct [8,9,10,11,12,13] or thermodynamic [14,15,16,17] methods. Recently, Zou showed that for a Ni-Zr system, the calculation efficiency for a direct voids method was not only higher than thermodynamic calculations, but also ultimately predicted similar *T_m_* values [18]. A simple example of a direct method is to heat a solid phase until the *T_m_* is reached, asdefined by a discontinuous change in density. This temperature is always higher than experimental *T_m_*. Compared to homogenous nucleation, the degree of superheating can be controlled through the addition of a heterogeneous interface, such as a void, which can reduce the free energy of melting [19].

Currently, the literature shows us several examples of studies using a direct void method. For example, Solca reported studies of a direct void method where it was shown that with the appropriate number of voids, the calculated *T_m_* could be taken to be the thermodynamic *T_m_* [20]. Agrawal extended this work to test the influence of the size, shape, and location of voids on the *T_m_* of argon and found the predicted *T_m_* is insensitive to the shape and size of the void but is, however, dependent on the total number of voids, and best agreed with experimental values with the number of voids in the plateau region of the percentage of voids versus the *T_m_* curve [11]. Many of these early MD studies have been directed toward atomic solids; therefore, some modifications and innovations are needed to compute the *T_m_* of cyanate esters. These monomers are excellent prototypical compounds for testing the feasibility of predicting melt behavior of thermosetting monomers, principally since there are dependable experimental data for these monomers (among others) to validate models available in the literature.

In this report, we examined four cyanate ester monomers (Figure 1) that were chosen on the basis of the available experimental data and modest structural similarities of the chemical structures. We applied an existing void method [20] to calculate *T_m_* and assess freedom of molecular motions and local ordering to characterize fundamental structure–property relationships. We found the void method gave reasonable estimates of *T_m_* for most monomers, although in some instances the plateau region in the percentage of voids versus predicted *T_m_* was narrow. While this result indicates the calculation of *T_m_* is somewhat limited in providing accurate melting temp values a priori, as a whole, the results shed light on how molecular structure may affect *T_m_* both computationally and experimentally and can be utilized to inspire the development of thermosetting monomers with optimal monomer melt properties for demanding applications.

## 2. Methods

### 2.1. Molecular Dynamics Simulations

MD simulations were performed to simulate the solid-to-liquid phase transition of cyanate ester monomers, calculate *T_m_*, and thereby appraise the quality of the molecular model relative to available experimental values (Figure 2). The Schrödinger Materials Science Suite [21] and Desmond [22] were used to for model construction, analysis, and simulation. Simulations were performed in the canonical ensemble with a constant number of atoms, pressure, and temperature (NPT) with a Nosé–Hoover thermostat [23,24] and the Martyna−Tobias−Klein barostat [25] unless otherwise noted. The OPLS3e forcefield was used for all molecules [26,27,28].

We examined four different cyanate ester monomers, with previously reported experimental melting temperatures: (1) 2′-(4-cyanatophenyl)propane (BADCy, *T_m_* = 355.15 K) [29], (2) 4,4′-(ethane-1,1-diyl)bis(cyanatobenzene) (LECy, *T_m_* = 302.15 K) [30], (3) tris(4-cyantophenyl)methylsilane (SiCy-3, *T_m_* = 390.62 K) [5], and (4) bis(4-cyanatophenyl)dimethylsilane, (SiMCy, *T_m_* = 333.05 K) [31]. The crystal structures of BADCy [32], LECy [30], SiCy-3 [5], and SiMCy [5] have also been previously published. The structures for BADCy (Refcode: TACHAG), SiCy-3 (Refcode: WAGWEJ), and SiMCy (Refcode: YERHAF) are available from the Cambridge Structural Database as .cif files [33]. For LECy, the crystallographic data were only available as unit cell vectors, so the .cif for BADCy was modified due to structural similarities. In this instance, a methyl group was deleted from each BADCy molecule in the unit cell, and subsequently a simulation cell was constructed from the edited structure. The supplied unit cells were replicated in all three directions to create supercells with 246 molecules (4 × 4 × 4) and 400 molecules (5 × 5 × 4). For 256-molecule ensembles (0% voids), LECy had the lowest number of atoms (8192), whereas SiCy-3 had the largest number of atoms (11,264). For larger systems with 400 molecules (0% voids), the number of atoms ranged from 12,800 to 17,600. The system sizes were chosen since they are large enough to ensure the boxes were greater than the minimum boundary condition (>20 nm), yet small enough to minimize the computational cost. For example, simulations containing 100,000 atoms take approximately 24 h to run on 8 cores. In view of this high computational cost of performing simulations with ~100,000 atoms, the use of smaller system sizes was particularly attractive.

The procedure to simulate the solid–liquid phase transition imitates the process of heating a crystalline solid from low temperature while tracking the change in volume of the system in tandem. To calculate *T_m_* from MD simulations, researchers have typically used the volume (or density) as a function of temperature [12,13,34,35,36,37,38]. Within this thermodynamic theory of melting, at the melting temperature (constant temperature and pressure), the solid–liquid equilibrium is defined as:(1)$${\frac{\partial G_{s}}{\partial P}}_{T_{m}}=V_{s}\neq V_{l}={\frac{\partial G_{l}}{\partial P}}_{T_{m}}$$
where *s* represents solid properties and *l* indicates liquid properties. For example, upon the first-order melting phase transition there is a discontinuous increase in volume.

We used a void method, put forward by Solca whereby voids are “added” into the system by deleting a certain percentage of monomers from the source structure [20]. Molecules were selected at random and deleted to prepare supercells with defect concentration ranges from 1 to 10% to allow us to assess the effect of voids on the quality of *T_m_* calculations with regard to experimental values. The 10% voids upper limit was selected to prevent mechanical destabilization and collapse of simulation cells, which has been previously attributed to an excessive number of defects, shear instability, or excessive vibrational motion [39]. An energy minimization step was used to generate optimized geometrical configurations and to provide molecule systems with realistic densities. MD simulations were run stepwise from 150 K specified increments until 600 K was reached. The simulation cells were subjected to an equilibration period of 20 ns at each temperature step, during which properties were elicited [40].

It should be mentioned that the heating rate influences thermodynamic outcomes [41]; therefore, initially two different heating rates (25 K/20 ns and 50 K/20 ns) were tested to observe the reproducibility of calculated *T_m_*.

### 2.2. Structural Characterization

#### 2.2.1. Orientational Order Parameter

The orientational order parameter was utilized to describe the solid-to-liquid phase transition. For a given configuration, this parameter is defined by:(2)$$OP=\left(\frac{1}{N}\right)\Sigma_{i=1,N}P_{2}\left(cos\theta_{i}\right)$$
where *cosθ_i_* is derived from the scalar product of a vector for the descriptor (principal moment of inertia) and the vector for the director, and *P*_2_(*cosθ_i_*) is the second-order Legendre polynomial, (3*cos*2*θ_i_* − 1)/2. The sum is calculated from all descriptor vectors defined for all molecules, and the order parameter is taken as the average value of the Legendre polynomial over all descriptors. *P*_2_ is used so that the order parameter does not distinguish between descriptor vectors in opposite directions [21]. The order parameter equals 1 when molecules have perfect alignment along the director, which signifies a crystalline lattice in the solid state. This value decreases as the system becomes disordered and approaches 0 when molecules have isotropic orientations with respect to the director, which indicates liquid-like behavior [37,42]. The procedure of the calculation consisted of running an annealing simulation below the *T_m_* (*T_m_* − 50 K) and above the *T_m_* (*T_m_* + 50 K). The simulation cells were subjected to an NVT Brownian dynamics stage for 1200 ps, an NVT MD stage for 1.2 ns at 10 K, and an NPT MD stage for 1.2 ns at the desired temperature to reach an equilibrium value using the MD Multistage Workflow module [43].

#### 2.2.2. Torsion

The torsion angle describing the orientation of the aromatic rings, relative to the bridging methyl group, was calculated using the Torsion Profile Analysis module. Trajectories were obtained in an analogous manner to the order parameter calculations. For all monomers, trajectory files for systems below and above the *T_m_* (*T_m_* − 50 K or *T_m_* + 50 K) were used as input for analysis over the simulation time. The SMARTS [44] pattern for the dihedral angle between the methyl carbon on bridging groups and aromatic carbons on the benzene ring was used to define the torsional angle.

## 3. Results and Discussion

### 3.1. Melting of Monomers with 0% Voids

The calculation of monomer melt properties relies on simulations that represent the existing physical systems and that are supported by empirical data. It is fundamental to our approach that simulating the systems should take less time compared to synthesizing and measuring properties experimentally. In an effort to reach a good compromise between simulation time and accuracy, we decided to apply a well-known void method [20] to our thermosetting monomer systems. Three parameters that may affect the calculated *T_m_* of simulated cyanate ester monomers were tested: system size, heating rate, and percentage voids. Simulations were performed as described above. The phase transition temperature of the simulated system was estimated by observing a discontinuity, to represent melting, or inflection, to represent a glass transition, from the temperature–volume curve. This is analogous to a *T_m_* calculation performed on a differential scanning calorimeter performed on themosetting monomers.

The effect of system size on the calculated *T_m_* was examined (Appendix A). We observed that increasing the number of molecules did not cause a significant change in the calculated *T_m_* except as noted in Appendix A, consistent with previous findings [20]. Based on this result, subsequent simulations were conducted with 256 molecules for efficiency. In addition, the optimal heating rate was tested on the 256-monomer systems. Simulations were run with stepwise temperature ramps of 25 K/20 ns and 50 K/20 ns (Appendix A). It was determined that there was no significant difference in the phase transition temperature, so the 50 K/20 ns (and 256 mol systems) was a good compromise between the simulation size and heating rate.

Figure 3 shows the evolution of volume–temperature curves (50 K/20 ns, 256 mol) for systems without voids. Due to the nature of the temperature increments, the *T_m_* was expected to be resolved within a 50 K temperature range, rather than a true discontinuity typical for a first-order phase transition. The intent was not to obtain an exact number, but rather a range sufficient to resolve a *T_m_* by 50 K, which would be useful for the purpose of ranking. Figure 3b shows the volume–temperature curve for LECy. The volume increases linearly up to 450 K, after which an abrupt increase in volume observed from 450 to 500 K. The *T_m_* is assigned to the onset of this discontinuity, which is evidence of the melting of a crystalline solid [35]. The volume–temperature graphs for BADCy and SiCy-3, shown in Figure 3a,c, do not exhibit a discontinuous *T_m_*. The discontinuous *T_m_* is likely not captured in our temperature range, so 600 K is taken as the calculated *T_m_*. For all systems with 0% voids, the predicted phase *T_m_s* are significantly greater than the experimental values. This finding is in good agreement with earlier observations of superheating for perfect crystalline lattices [13,45,46].

The calculated *Tms* for all of the simulated monomer systems were overestimated, which was expected and has previously been attributed to the high-free-energy barrier to melting [13]. The barrier to melting for simulations without voids is largely on account that homogenous nucleation is the sole mechanism to initiate melting, which is determined by the probability of spontaneously forming liquid-like droplets in the solid phase [13]. In the absence of voids, another important consideration is that melting is a fundamentally a kinetic phenomenon, and varying the heating rate is known to affect the observed *T_m_* [47]. Heating rates achieved in MD simulations are intrinsically much faster than heating rates accessible for experimental systems. In this context, significant superheating was observed to occur since molecular motions are hindered below the *T_m_* at short timescales accessible in MD simulations.

### 3.2. Dependence of Monomer Melting Temperature on Percentage of Voids

The following part of this study involved removal of 1 to 10% of molecules from simulation cells. This was to assess the effect of the void percent on the quality of the *T_m_* calculation with regard to experimental values and to observe the monomer melt behavior on an atomistic scale. The 10% voids upper limit was established to prevent mechanical instability and collapse of the simulated crystalline solid into an amorphous solid [20]. The evolution of volume as a function of stepwise heating for monomers is shown in Appendix A. The variation in the calculated *T_m_* as a function of percentage of total number of molecules removed from simulation cells is shown in Figure 4a. The most striking and immediate observation for the plots is the decrease in the predicted *T_m_* with an increase in percent voids, complementing previous studies [11,20]. The lower *T_m_* reflects a decrease in the superheating effect in the direct heating to melting on account of a decrease in the free-energy barrier to the formation of a solid–liquid interface to nucleate melting. In general, monomers LECy and SiMCy have the lowest *T_m_s*, BADCy an intermediate value, whereas SiCy-3 has the highest *T_m_*.

While the voids were confirmed to be necessary to nucleate melting of the simulated thermosetting monomers, there were some issues with the application of the void method to these polyatomic monomers. In the traditional void method, the estimated *T_m_* has been shown to reach a constant value that is independent of the percentage. This was not observed in all thermosetting monomers simulated; therefore, an important consideration is the discrepancy from the traditional voids method, with regard to the absence of a flat region. It is evident that LECy and SiMCy are in best agreement with the conventional void method as they reach an appreciable plateau from 6 to 10% voids (Figure 4a). In the case of BPACy and SiCy-3, the systems do reach a narrow plateau where *T_m_* is independent of percent voids; however, it spans from 9 to 10% and is more narrow than expected. The authors recognize that an important consideration regarding the calculations is the lack of a theoretical foundation in the assumption that the thermodynamic *Tm* for a crystal coincides with a plateau region for a crystal lattice with a particular void size. Furthermore, a meaningful variable that may affect the *T_m_* values in the plateau region is the possible inaccuracy related with superheating due to the rapid heating of simulated monomers. On the other hand, the current study does support the observations of Solca that the *T_m_* decreases with an increase in the percentage of voids, then remains constant in a “plateau region” that corresponds to the thermodynamic *T_m_* [20]. Alvares used the voids method to calculate *T_m_* for CaO, and the plateau value was reported to span from 9 to 27% voids [40]. For Alavi, the plateau spanned from 5 to 10% voids, despite the similar approach [19]. These findings emphasize the difficulty in predicting, a priori, (and calculating computationally) the *T_m_* of monomers. A challenge that becomes formidable when targeting additional constraints on reactivity and performance to satisfy composite applications.

Appendix A compared the accuracy of calculated *T_m_s* against the experimental values, as a function of the percentage of voids. Most of the calculated values are well above the experimental *T_m_* partly due to the lack of heterogeneous nucleation and also due to the measurement technique (heating rate). Of note is the good agreement of calculated *T_m_s* with experimental values from 9 to 10% voids for most monomer systems. Since our simulations were run in 50 K steps, we expected the technique would be capable of ranking modeled systems, rather than precise numeric agreement. With this consideration, the 9% void systems were established for the comparison analysis between simulated monomer systems and are summarized in Table 1 and Figure 4b. Additionally, there is good agreement of the results on BPACy, LECy, and SiCy-3, and to a lesser degree SiMCy, with the experiment.

A point to note from Appendix A is that the *T_m_* of SiMCy was most similar to the experimental value from 3 to 5% voids, but the accuracy diminishes further with increasing percentage of voids. This behavior has been observed elsewhere and is related to the mechanical instability and collapse of the simulated crystalline solid into an amorphous solid [20]. While this was unexpected when considering results from other monomers, when the volume versus temperature curves for SiMCy were plotted (Figure 5a), important differences between simulated behavior became evident. It can be seen that simulations of SiMCy with 9% and 10% voids exhibited an inflection, which suggests a glass transition to a rubbery regime. The appearance of the T_g_ suggests the incorporation of high void percentages may cause the collapse of the solid lattice into an amorphous solid at the start of the MD simulation because the system is unable to maintain solid configuration [48]. It should be mentioned for the simulated systems with 256 molecules that the voids are located relatively close to each other. In the instance of high percent voids, these interfaces may overlap, transforming the solid–liquid phase into supercooling amorphous liquid. Considering the effect of size, it can be seen that for the system with 400 molecules, all the simulations exhibited a discontinuous solid-to-liquid melting phase transition (Figure 5b). In a comparison of the four monomers with 9% voids, it can be seen that the correct rank ordering is retained.

**Table 1 polymers-14-01219-t001:** The melting points of cyanate ester monomers with 9% voids compared to previously reported literature values.

Monomer	Calculated *T_m_* (K)	Experimental *T_m_* (K)	Reference	Percent Error (%)
BADCy	350	355.15	[49]	2
LECy	300	302.15	[30]	1
SiCy-3	400	391.3	[5]	3
SiMCy	300	333.05	[31]	10

### 3.3. Structural Characterization

Following the examination and validation of system construction with *T_m_* calculations, the structural parameters were also explored. Order parameter studies were performed to determine the possible correlation of local short-ranged ordering to *T_m_* and observe monomer melt behavior on the molecular level. The order parameter in MS suite was utilized and applied to monomer systems with 9% voids. The calculations involved the use of simulation cells slightly below (*T_m_* − 50 K) and above (*T_m_* + 50 K) the *T_m_*. The low- and high-temperature order parameters for each class of monomer were repeatable across multiple simulations and are summarized in Table 2, and snapshots of this order-to-disorder transition are shown in Appendix A. The simulated values were expected to be at a maximum for monomers with high *T_m_s*, and at a minimum for monomers with comparatively lower *T_m_s*.

The orientational order parameters for the monomers below the *T_m_* vary from ~0.6 to ~0.3 and are ranked from highest to lowest in the following order: SiCy-3 > BADCy > LECy > SiMCy. SiCy-3 has the highest order parameter and the highest experimental *T_m_* of the monomers characterized in this study. Analysis shows a comparable ordering between BADCy and LECy systems, with BADCy being slightly higher. It is well-known that the BPA-derived cyanate ester monomer with two methyl groups on the bridgehead tend to have increased rigidity. Compared to BADCy, LECy has one less methyl group on the bridgehead group, resulting in an increase in flexibility and decrease in close contact ordering (Figure 6). The correlation between order parameter and *T_m_* seen here lends support to the hypothesis that an increase in ordering between close contacts is the driving factor in the melting temperatures.

However, it was surprising the same trend does not hold with SiMCy, as it has the lowest order parameter, yet the second lowest *T_m_*. This result may be rationalized by the monomer chemical structure and that the increased flexibility from the silicon substituted bridgehead increases isotropy in the ordering, resulting in the lowest order parameter value. Work by Guenthner has shown that the rotational barriers experienced by the silicon containing monomer is low compared to BADCy (quaternary carbon) due to “extra” conformational freedom of SiMCy due to the quaternary silicon group [5]. For SiMCy, the two phenyl rings are not “locked in” and can occupy a variety of twist angles, which may rationalize the low-order parameter. It should also be mentioned that the low-order parameter value signifies the presence of amorphous domains in SiMCy simulation cells, which is in agreement with observation of a T_g_ for SiMCy systems with 9 to 10% voids. When considering the previous section where the calculated *T_m_* value (9% voids) was compared to the experimental *T_m_* value, the presence of amorphous domains, rather than the crystal structure, may result in the lack of congruous trends formerly observed with molecular simulations using voids method (underpredicted *T_m_*).

For simulations run above *T_m_*, all monomers exhibited observable drops in order parameter that approached zero over time, which was expected. As seen in Figure 7, BADCy was observed to reach an equilibrium value of 0.09, indicating the crystalline lattice structure completely dematerializes [50]. Similar trends were observed for the other monomers (Appendix A). These observations indicate an order-to-disorder transition as a result of the melting process, specifically an increase in mobility and decrease in associations [51,52].

To further investigate structure–property relationships and examine possible correlations between chemical structure (freedom of motion) and monomer *T_m_*, Torsion Profile Analysis was performed on monomer systems. The module allows for the investigation of a property that is not readily accessible experimentally: phenyl ring torsion of monomer melts. It was hypothesized that the difference in *T_m_* between monomers was influenced by the chemical structure and assumed flexibility of the bridgehead group [5]. Furthermore, phenylene flips would indicate the solid-to-liquid phase transition, with high-melting monomers having a narrow distribution of dihedral angles, indicating high rigidity. Conversely, it was expected that low-melting monomers would have broader distributions of dihedral angles, indicating increased monomer flexibility.

In order to examine this, equilibrated systems slightly below (*T_m_* − 50 K) and above (*T_m_* + 50 K) the *T_m_* were examined. For most systems, 9% voids was used. For SiMCy, 7% voids was used, since the system maintained a crystalline structure prior to melting, as evidenced by a first-order solid-to-liquid phase transition. Figure 8 shows examples of changes in the dihedral distribution (phenyl group orientation compared to methyl bridging group) upon the solid–liquid phase transitions. In the case of BADCy, below the *T_m_*, the mean values fluctuate from around −174 to −154°; −124° and −94 to −84°; −54° and −30 to 25°; 55° and 85 to 95°; and 125° and 155 to 175°. The peak positions of the BADCy dihedral angle shift as the temperature is increased from 300 to 450 K. The pronounced peaks in the distributions merge and range from −175 to −125°, −95 to 55°, and 95 to 175°. Not only do the peak widths in the distribution increase, but the peak heights also decrease upon melting, signifying a decrease in the barrier to motion as the relative populations of conformers broaden [53]. The torsion of phenyl rings with respect to the two methyl groups at the bridge is able to assume a broad range of conformations, and the ring acquires more freedom to rotate. A similar trend is seen with LECy (Figure 8b). Below the *T_m_*, the torsion has maxima at −153 to −143°, −113 to −64°, −34 to −24°, 15 to 35°, 65 to 75°, 94°, 114°, and 144 to 154°. Above the *T_m_*, the probability greatly changes, and the maxima occur from −142 to −123°, −63 to −24°, 26 to 56°, and 105 to 145°. Compared to BADCy, it can be seen that LECy has more broad peaks below the *T_m_*. This broadening indicates a lower barrier to motion for LECy compared to BADCy, which would be expected on the basis of the decrease in rotational steric hinderance upon replacing one methyl bridging group with a hydrogen atom. Ultimately, the lower barrier to motion results in a concomitant decrease in *T_m_*_,_ and the expectedly lower order parameter seen for LECy in the previous section. However, when considering the distribution of dihedral angles for all of the monomers in the data set, important differences between monomer freedom of motion become apparent. Figure 8 shows the broadest peaks in the melt (above the *T_m_*) are in the order of: BADCy > SiMCy > SiCy-3 > LECy. However, the *T_m_* values occur in the following order: SiCy-3 > BADCy > SiMCy > LECy. While the computed results appear contradictory to previously reported experimental value, this suggests that the barrier to motion (internal entropy) is not the sole contributor to the *T_m_*, and other factors should be considered, such as interactions with neighboring molecules (intermolecular interactions). These results align with those of Ghiassi, who used X-ray crystallography and thermal and computational simulations to study the structure–property relationships that determine the melting characteristics for a series of cyanate ester monomers. The authors similarly concluded the expected correlation between restricted motion and entropy of melting (melting characteristics) was not observed for the examined monomers [7].

The dihedrals between aromatic rings and a bridging methyl group are of particular interest because they define the phenyl ring positions in the molecule. In order to explore this further, the dihedral angle was plotted as a function of time during equilibration for BADCy (Figure 9). The sequence of images shows the time-dependent distributions of dihedral angles that demonstrate the phase transition of BADCy monomers upon heating above the *T_m_*. In the crystal state the torsion angles were relatively stagnant and distributed without deviation along a gradient, as expected. At 450 K, a fluctuation in the dihedral angle of phenyl group orientation compared to methyl bridging group suggests ring flips occur, likely due to the increase in thermal energy when the melting temperature is approached. The same trend can be seen for all systems (Appendix A), which show fluctuating dihedral angles upon melting, in accordance with the increase in thermal energy and freedom of motion.

## 4. Conclusions

This research was aimed at studying the effect of chemical structure on the melting behavior of cyanate ester thermosetting monomers and investigating the performance of the voids method introduced by Solca in capturing the melting temperature [20]. MD simulations were carried out on the crystalline monomers with void percentages up to 10%. A computational framework and a compared analysis of simulated *T_m_* is presented and validated through comparison to previously reported empirical data. Additionally, the torsion and orientational order parameter were used to examine the monomers’ freedom of motion in order to determine structure–property relationships. It was found that for crystal lattices with up to 400 molecules, 9% voids provided the best estimate of *T_m_* for most systems. The orientational order parameter and torsion simulations gave insight on the local short-ranged ordering of monomers and monomer flexibility. The monomers with lower melting temperatures tended to have lower orientational order parameters than monomers with higher values, as expected [54], based on molecular symmetry and packing. Furthermore, these results, in tandem with torsion simulations, indicate that the internal barrier to motion is not the sole contributor to *T_m_*, and intermolecular interactions must also be considered. As a whole, the results shed light on how the chemical structure of cyanate ester monomers may affect *T_m_*, which was both computationally and experimentally determined, and can be utilized to inspire the development of thermosetting monomers with optimal monomer melt properties for demanding applications.

## Figures and Tables

**Figure 1 polymers-14-01219-f001:**
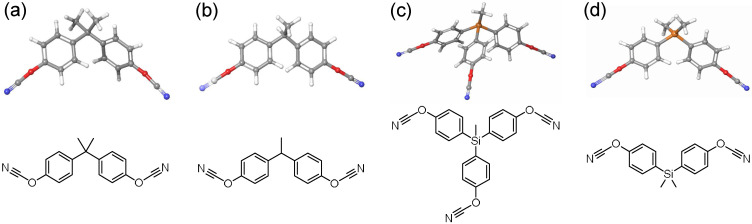
Chemical structures and three-dimensional illustrations of monomers: (**a**) 2′-(4-cyanatophenyl)propane (BADCy), (**b**) 4,4′-(ethane-1,1-diyl)bis(cyanatobenzene) (LECy), (**c**) tris(4-cyantophenyl)methylsilane (SiCy-3), and (**d**) bis(4-cyanatophenyl)dimethylsilane (SiMCy).

**Figure 2 polymers-14-01219-f002:**
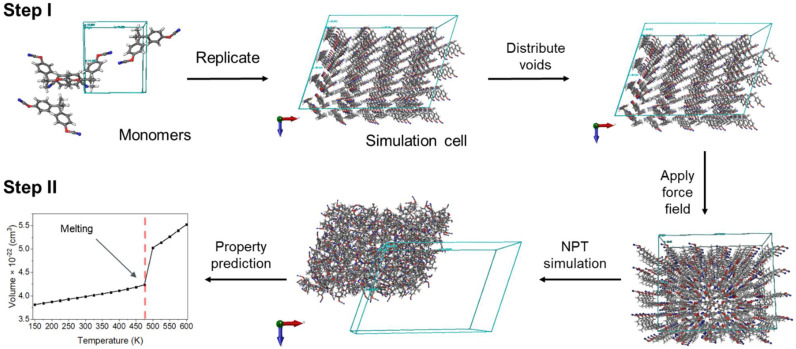
Molecular Dynamics (MD) workflow of melting temperature simulations.

**Figure 3 polymers-14-01219-f003:**
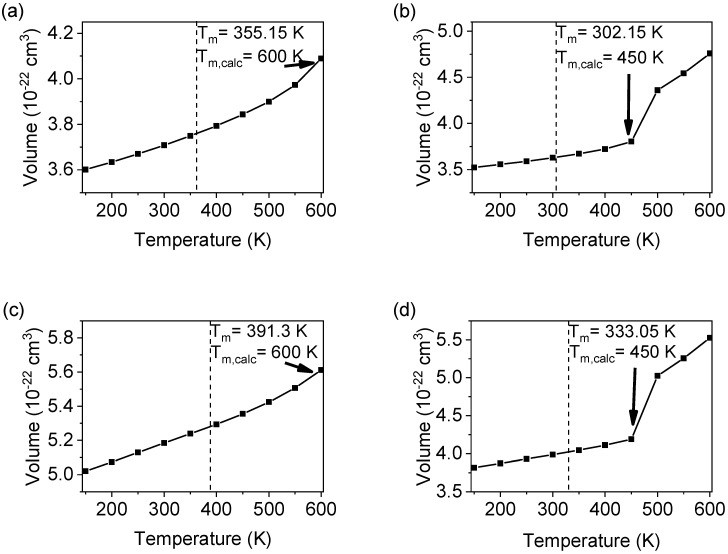
The volume as a function of temperature for monomers: (**a**) BADCy, (**b**) LECy, (**c**) SiCy-3, and (**d**) SiMCy with 0% voids. The calculated melting temperature (*T_m,calc_*) is represented by the discontinuity or inflection (arrow). The results were compared to previously reported experimental melting temperatures (dotted line).

**Figure 4 polymers-14-01219-f004:**
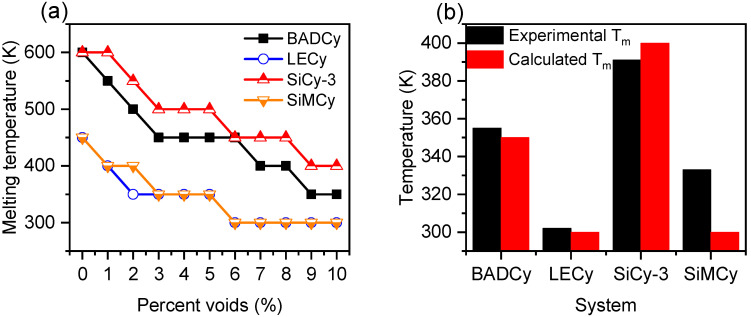
(**a**) The calculated melting temperature as a function of percentage voids (in % of the total monomers removed from simulation cell) and (**b**) calculated *T_m_* at 9% voids compared to experimental *T_m_*.

**Figure 5 polymers-14-01219-f005:**
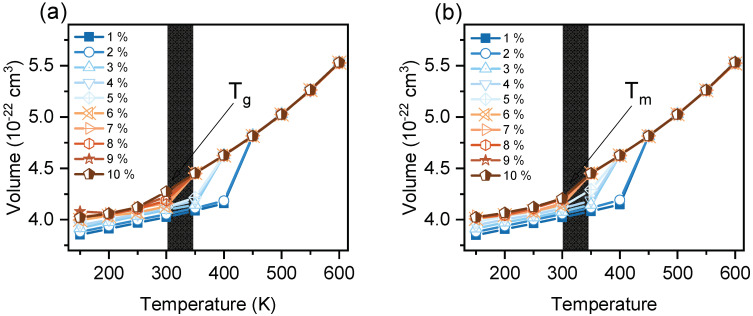
The evolution of volume as a function of temperature for SiMCy with: (**a**) 256 and (**b**) 400 molecules.

**Figure 6 polymers-14-01219-f006:**
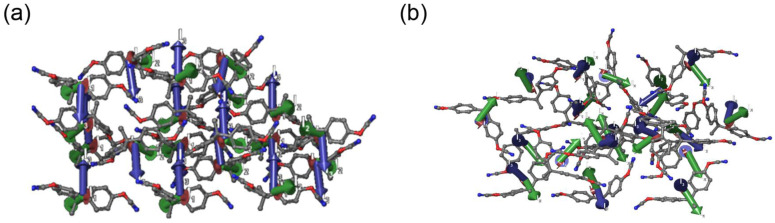
Snapshots of (**a**) BADCy and (**b**) LECy, highlighting the ordering of close contacts. The principal moments of inertia are displayed as a set of orthogonal tubes.

**Figure 7 polymers-14-01219-f007:**
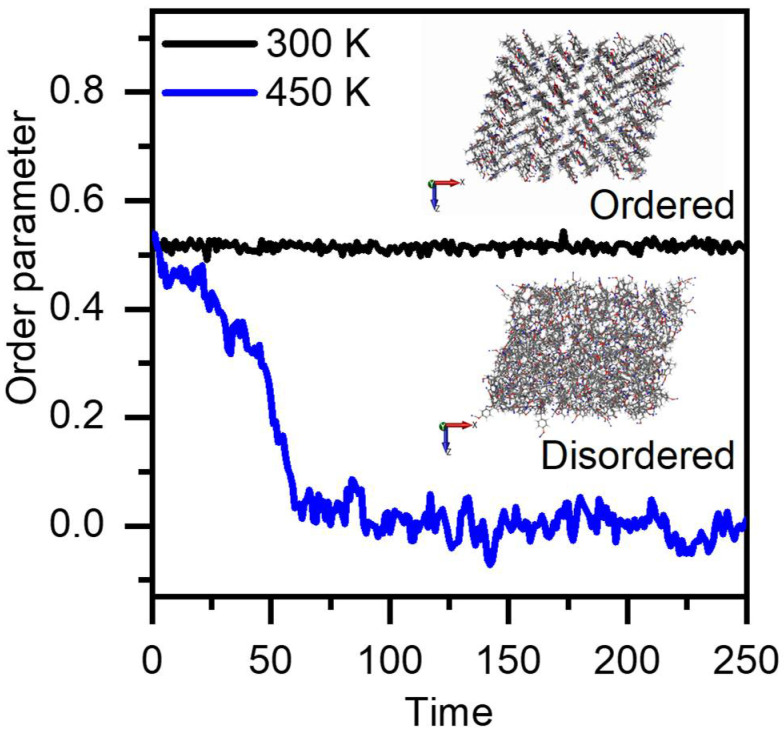
The orientational order parameter for BADCy 50 K below the calculated melting temperature (black line) and 50 K above the calculated melting temperature (blue line). Included is a snapshot of monomers in the ordered state below the calculated melting temperature and disordered state above the calculated melting temperature.

**Figure 8 polymers-14-01219-f008:**
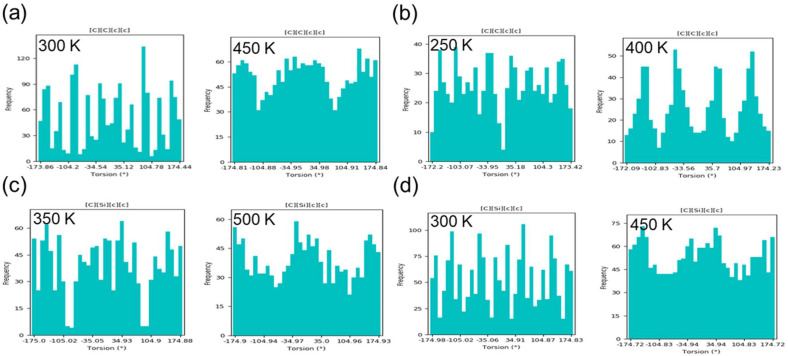
Histogram plots to examine the distribution angle values for an individual torsion for: (**a**) BADCy, (**b**) LECy, (**c**) SiCy-3, and (**d**) SiMCy.

**Figure 9 polymers-14-01219-f009:**
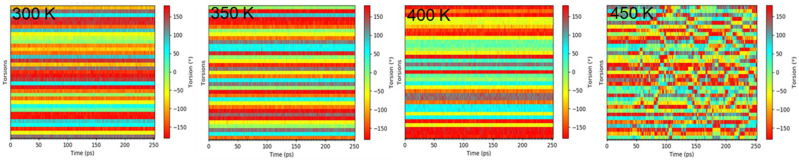
Display of angles for each individual torsion as a bar plot for the simulation of a time range with colors denoting the size of the angle for BADCy.

**Table 2 polymers-14-01219-t002:** The orientational order parameter values for monomers: 50 K below the calculated melting temperature and 50 K above the calculated melting temperature.

Monomer	Order Parameter below *T_m_*	Order Parameter above *T_m_*
BADCy	0.515 (300 K)	0.09 (450 K)
LECy	0.41 (250 K)	0.06 (400 K)
SiCy-3	0.61 (350 K)	0.06 (500 K)
SiMCy	0.24 (300 K)	0.03 (450 K)

## Data Availability

The data presented in this study are available in the article and Appendix A.

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
