# Peer review of "A Molecular Dynamics Study of Cyanate Ester Monomer Melt Properties"

_polymers, 2022, doi:10.3390/polym14061219_

Round 1
Reviewer 1 Report
This work predicted the melting temperature and melt properties of thermosetting monomers used in aerospace applications by using molecular dynamics method. The results are of some interest, however, there are still some problems that the authors should be properly addressed:
- In Figure 3he temperature interval is too large, resulting in a large error in Tm, calc. Besides, it is difficult to judge that the Tm, calc are both 600 K in Figure 3 a & c.
- How much pressure was used in the simulation process under the NPT ensemble? How does density affect the melting point of this system?
- Variables should use italics, such as Tm..
Author Response
Response to Reviewer 1 Comments polymers-1613674: The authors would like to thank Reviewer 1 for their instructive critiques to improve the quality and presentation of this manuscript. We found your comments helpful and appropriate and clarify our revisions herein. We present to you our item-by-item response and look forward to your support for publication.
REVIEWER 1 COMMENTS:
Comments and Suggestions for Authors
This work predicted the melting temperature and melt properties of thermosetting monomers used in aerospace applications by using molecular dynamics method. The results are of some interest, however, there are still some problems that the authors should be properly addressed:
- In Figure 3he temperature interval is too large, resulting in a large error in Tm, calc. (We elected to run our simulations with large temperature steps of 50 K or 25 K to minimize the time to complete each simulation. For example, when heating a monomer system each individual temperature step took approximately 2- 4 hours. We agree optimization through extending this research in a broad array of temperature steps is an appropriate next step.) Besides, it is difficult to judge that the Tm, calc are both 600 K in Figure 3 a & c. (REVISED AS SUGGESTED)
- How much pressure was used in the simulation process under the NPT ensemble? (Previous studies using same force field showed densities at 1 atm within 2.18% of experiment on average (Afzal et al)). How does density affect the melting point of this system? (While not measured in this study, the melting temperature would be expected to go down with increasing pressure and density)
- Variables should use italics, such as Tm. (REVISED AS SUGGESTED)

Reviewer 2 Report
This paper by Haber et al. applied the existing voids method with molecular dynamics simulations to the prediction of melting temperature of four cyanate ester monomers. A few order parameters were used to understand the monomers’ motion and structure-property relationships. I think the topic of this work is of good importance and interest, but there are several points below needed to be further addressed before I can recommend its publication to Polymers.
Major comments:
- What is the logic of choosing those four monomers? Is it because the melting temperatures of these four monomers are available? Are there any other monomers’ Tm also available?
- On page 3 line 123, is there a reason why these particular numbers (256, 400) of molecules are chosen?
- Would the void size have any effect on the melting temperature? For example, if keeping the fraction of void the same, but increasing the size of every unit of void (currently 1, can increase to 2 or 4), how would the results look like? Basically, by increasing the size of the void, it equals to one forces a certain number of overlapping of individual voids in the settings of the current manuscript.
- For Figure 3, is there any explanation on why with 0% voids the simulated Tm of BADCy and SiCy-3 are even much higher (>600K) than the experimental values?
- Similarly, is there any explanation on why the Tm from simulations of LECy and SiMCy are both 450K with 0% voids?
- For Figure 4a, the authors should rationalize by discussing more of the choice of stopping at a void percentage of 10%, where BADCy and SiCy-3 are not entirely clear whether they are plateau in terms of Tm as a function of the percentage of voids.
- For Figure 7, I wonder if the authors perform multiple simulations, with the same void percentage but different initial void configurations, how large would the results be different across different simulations? This would be another validation of the convergence of the performed simulations.
- The authors claim that “systems with 9% voids provided the best estimate of Tm for most systems”, but if I understand correctly, this is the conclusion for the 256-molecule system. What would be the optimal void percentage for larger systems? Since as the author pointed out, smaller systems have more serve size-dependent issues.
- The authors should discuss more on the efficiency of the computational simulations. For example, how much would it cost for simulating a 400-molecule or even larger system? Is it on orders of magnitude cheaper and faster than the experimental characterization? How in quantitative numbers?
- One of my major concerns is how much more value can this work add to the current development of thermosetting monomers for demanding applications as the authors claimed. It seems that the Tm cannot be predicted accurately in a quantitative manner anyway. The authors should give more careful and insightful high-level discussions.
Minor comments:
- On page 10 line 347, the authors claim that “Similar trends were observed for the other monomers”. Currently, I didn’t see any figures for this, the figure/data for this claim need to be shown.
Author Response
Response to Reviewer 2 Comments polymers-1613674: The authors would like to thank Reviewer 2 for their instructive critiques to improve the quality and presentation of this manuscript. We found your comments helpful and appropriate and clarify our revisions herein. We present to you our item-by-item response and look forward to your support for publication.
Major comments:
- What is the logic of choosing those four monomers? Is it because the melting temperatures of these four monomers are available? Are there any other monomers’ Tm also available? (REVISED AS SUGGESTED)
- On page 3 line 123, is there a reason why these particular numbers (256, 400) of molecules are chosen? (REVISED AS SUGGESTED)
- Would the void size have any effect on the melting temperature? For example, if keeping the fraction of void the same, but increasing the size of every unit of void (currently 1, can increase to 2 or 4), how would the results look like? Basically, by increasing the size of the void, it equals to one forces a certain number of overlapping of individual voids in the settings of the current manuscript. (REVISED AS SUGGESTED)
- For Figure 3, is there any explanation on why with 0% voids the simulated Tm of BADCy and SiCy-3 are even much higher (>600K) than the experimental values? (REVISED AS SUGGESTED)
- Similarly, is there any explanation on why the Tm from simulations of LECy and SiMCy are both 450K with 0% voids? (REVISED AS SUGGESTED)
- For Figure 4a, the authors should rationalize by discussing more of the choice of stopping at a void percentage of 10%, where BADCy and SiCy-3 are not entirely clear whether they are plateau in terms of Tm as a function of the percentage of voids. (REVISED AS SUGGESTED)
- For Figure 7, I wonder if the authors perform multiple simulations, with the same void percentage but different initial void configurations, how large would the results be different across different simulations? This would be another validation of the convergence of the performed simulations. (REVISED AS SUGGESTED)
- The authors claim that “systems with 9% voids provided the best estimate of Tm for most systems”, but if I understand correctly, this is the conclusion for the 256-molecule system. What would be the optimal void percentage for larger systems? Since as the author pointed out, smaller systems have more serve size-dependent issues. (REVISED AS SUGGESTED)
- The authors should discuss more on the efficiency of the computational simulations. For example, how much would it cost for simulating a 400-molecule or even larger system? Is it on orders of magnitude cheaper and faster than the experimental characterization? How in quantitative numbers? (REVISED AS SUGGESTED)
- One of my major concerns is how much more value can this work add to the current development of thermosetting monomers for demanding applications as the authors claimed. It seems that the Tm cannot be predicted accurately in a quantitative manner anyway. The authors should give more careful and insightful high-level discussions. (REVISED AS SUGGESTED)
Minor comments:
- On page 10 line 347, the authors claim that “Similar trends were observed for the other monomers”. Currently, I didn’t see any figures for this, the figure/data for this claim need to be shown. (REVISED AS SUGGESTED)

Reviewer 3 Report
Abstract and conclusion are long-winded.
Author Response
Response to Reviewer 3 Comments polymers-1613674: The authors would like to thank Reviewer 3 for their instructive critiques to improve the quality and presentation of this manuscript. We found your comments helpful and appropriate and clarify our revisions herein. We present to you our item-by-item response and look forward to your support for publication.
- Abstract and conclusion are long-winded. – (We appreciate this comment but uncertain how to specifically address the concern – we will be mindful of this comment with future submissions)

Round 2
Reviewer 1 Report
Accept in present form
Reviewer 2 Report
The revised manuscript has largely improved through reading it. However, I found it strange by looking at the item-by-item response to my previous comments. There is nothing but one phrase: (REVISED AS SUGGESTED). Even if the authors have revised the manuscript accordingly, it would still be necessary to respond item by item in the response letter, especially some of my previous comments were questions waiting for detailed explanations, and I don't see any proper answers show up (even if they are embedded in the new version of manuscript).
Reviewer 3 Report
This manuscript can be published.